# Impact of New Energy Industry Agglomeration on Green Innovation Efficiency—Based on the Regulative Effect of Green Finance

**Yiding Wu [1,2,3,*] and Jingfei Song [1]**

[1]   School of Economics and Management, Jiangxi University of Science and Technology, Ganzhou 341000, China; 6120220006@mail.jxust.edu.cn
[2]   Ganjiang Innovation Research Institute, Chinese Academy of Sciences, Ganzhou 341000, China
[3]   Ganzhou Research Institute, Jiangxi University of Finance and Economics, Ganzhou 341000, China
*   Correspondence: wid0410@163.com

**Abstract:** With the implementation of China's innovation-driven high-quality economic development strategy, green and innovation are already the key factors of economic development. Therefore, developing green industry and improving regional green innovation have attracted wide attention and are of great significance to the sustainable development of China's economy. Therefore, starting from China's provincial panel from 2012 to 2021, this paper first uses the super-efficiency relaxation data envelopment analysis model (Super-SBM) to estimate green innovation efficiency (*GI*) and then uses the location entropy to measure the regional agglomeration level of the new energy industry (*agg*). Then, the generalized estimation of moments (GMM) model is used to explore the impact of *agg* on *GI* and verify the regulatory mechanism of green finance (*GF*). The results are as follows: (1) *agg* presents a distribution of "the highest in the eastern region, followed by the central region, and the lowest in the western region", (2) *agg* can facilitate the improvement of *GI*, and in accordance with the threshold model, moderate *GF* will further amplify this effect. Therefore, the state and government should further promote the green finance policy, guide new energy enterprises to gather and contribute to the sustainable development of China's economy.

**Keywords:** new energy industry agglomeration; green innovation efficiency; green finance; dynamic system GMM; thermal map

## 1. Introduction

Since the reform and opening up, China's economy has been growing at a high growth rate [1]. However, the resource- and factor-driven development model of "high input, high consumption and high emission" leads to resource constraints and the continuous intensification of environmental problems [2,3]. In this context, innovation-driven development is an important strategy for China's sustainable economic development [4]. This means that China's economy will no longer mainly rely on the low-cost advantage of labor, resources and the environment but will magnify the productivity of various production factors by using the multiplier effect of innovation, so as to achieve efficient economic growth [5]. During China's 14th Five-Year Plan, green and innovation became key factors for economic development. At the micro level, green innovation improves the utilization efficiency of natural resources and cuts down the dependence on the natural environment through the R&D and promotion of green products and processes [6,7]. At the macro level, the coordinated development of the economy, resources and the environment is promoted through innovation-driven development, so as to achieve green economic development [8]. Based on this, green innovation efficiency (*GI*) has become an important indicator for measuring regional sustainable development. In accordance with the China Green Patent Statistical Report 2023 disclosed by the State Intellectual Property Rights of China, the quantity of

green patents authorized in China accounts for 36.8% of the world, but the quantity of green patents authorized in each province shows an unbalanced trend [9]. Therefore, it is of extremely practical significance to improve *GI* for the sustainable development of China's economy.

Schumpeter's early research proves that there exists a mutually reinforcing relationship between the spatial agglomeration of industry and technological innovation. Agglomeration economy guides development factors and economic activities to gather in a specific space through increasing returns to scale, imperfect competition and knowledge and technology spillovers and has an impact on regional economic development [10,11]. By virtue of its knowledge- and technology-intensive characteristics, high-tech industry agglomeration relies on innovation to affect the development of green and innovation. The implementation of China's "dual carbon" target has promoted the rapid development of the new energy industry. This industry has obvious green and high-tech characteristics, and its development contributes to the improvement of China's overall and regional green innovation efficiency. At the current stage, China's new energy industry has met the five conditions for cluster formation, and this is the future direction of the industry's development [12]. However, as a high-tech industry, its technological innovation is easily limited by financing constraints, which are determined by the characteristics of high investment, high risk and a long recovery cycle of technological innovation in this industry. Therefore, in order to support the development of this industry, the Chinese government has introduced a number of financial policies, such as government subsidies and tax incentives. Among these measures, green finance policy is vital for the technological innovation of industries. The policy is a financial innovation that links the environment and the economy; it mainly restricts banks and other financial institutions from distributing funds to heavily polluting enterprises and increases financial support for green industries to promote industrial development [13]. Under the above background, can *agg* promote *GI*? How does the level of *GF* affect the relationship between the two variables? In view of the above problems, this paper takes green innovation efficiency as the research object; constructs an evaluation index system of the new energy industry agglomeration (*agg*), green finance index (*GF*) and *GI*; and introduces a dynamic panel regression model, namely the generalized method of moments (GMM) system, to study the impact of *agg* on *GI* and explore the regulating effect of *GF*. This paper is of great practical significance for improving the efficiency of regional green innovation, evaluating the implementation effect of green finance policies and promoting high-quality regional development.

The sections following the introduction are structured as follows: The second section presents the literature review. It is mainly constructed from two perspectives: industrial agglomeration and green innovation efficiency, green finance policy and new energy industry development. The third section presents the theoretical analysis and research hypotheses. The fourth part is the construction of the econometric model and the selection of variables and data. The fifth part is the analysis of empirical results and robustness test. The conclusions, suggestions and limitations of this paper are all mentioned in the sixth part.

## 2. Literature Review

### 2.1. Industrial Agglomeration and Green Innovation Efficiency

As resource constraints and environmental pressure continue to intensify, academia has begun to pay attention to green innovation. As for the concept of green innovation, there are the following three explanations: the first, is a reduction in environmental impact [14]; the second is the introduction of environmental performance [15–17]; the third is environmental innovation or environmental performance improvement. The above definition of green innovation not only describes its "innovation" attribute, but also describes its "environmental benefit" attribute. Green innovation is an indicator that estimates green development level, and environmental benefits are incorporated into its input–output. At present, the studies on *GI* concentrate on two aspects, including measurement, analysis and

influence factors of this indicator. In the first aspect, the data envelopment analysis model (DEA) and stochastic frontier analysis model (SFA) multi-input multi-output models are mainly used for measurement, but relevant environmental variables, such as green patent output, are added to the selection of input–output indicators [18–20]. In the second aspect, market, technology, environmental regulation, economic development and industrial agglomeration have been confirmed to have a significant impact on the efficiency of green innovation [21].

However, regarding the impact of industrial agglomeration on green innovation efficiency, there is no consensus. One view holds that Marshall externalities, Jacob externalities and Porter externalities are the mechanisms through which industrial agglomeration can promote the efficiency of green innovation. Industrial agglomeration is a manifestation of a high geographical concentration of enterprises, which will reduce the production cost of enterprises through externalities and then have an impact on the regional economy [22]. Chinese scholars have also tested this phenomenon; for example, Zhang and Shen [23] confirmed that industrial agglomeration would contribute to regional innovation efficiency through the above three externalities. Wang et al. [24] made use of the spatial econometric method to confirm the first view. Meanwhile, Wang and He [25] also adopted the spatial Durbin model, which not only verified the same result, but also tested the spatial spillover effect of industrial agglomeration. The second view is that industrial agglomeration will restrain the positive impact of industrial agglomeration on green innovation efficiency because the cost of competition will be increased by agglomeration, which will weaken the positive influence of industrial agglomeration on green innovation efficiency [26,27]. Leeuw et al. [28] discussed the inhibition of industrial agglomeration on *GI* in large cities of EU. Some Chinese scholars have also taken advantage of spatial metrological model to verify that the promotion of *GI* will be influenced by specialized agglomeration in the industrial agglomeration. Moreover, this phenomenon also appeared in the industrial agglomeration of high-tech industries [29,30]. Another view is that the impact of industrial agglomeration on *GI* is nonlinear, which means it is inverted U-shaped or N-shaped [31,32]. Wang et al. [33] further found that an inverted U-shape appeared in studying the impact of different agglomeration modes on urban green TFP. The above results are also applied in the tourism industry [34].

## 2.2. Green Finance Policy and New Energy Industry Development

The new energy industry is playing a crucial role in the global social and economic transformation. The development, production, consumption and challenges of renewable energy are explored by scholars, and it is suggested that the exploitation of this industry will contribute to the economic growth of the United States, Canada and Mexico [35]. Furthermore, the industry, as a green and clean industry, will exert a significant influence on *GI* by its agglomeration effect. However, there are few studies on *agg* in the academic circle. The measurement, influencing factors and economic consequences of agglomeration of the new energy industry are studied by Chinese scholars. Since China does not have statistics on this industry currently, the measurement methods mainly include output value substitution and the improved HHI index [36–39]. As for the influencing factors of *agg*, Qiu et al. [40] selected the environmental policies of the EU as objects of study, quantified these policies and found that they can greatly affect *agg*. The economic consequences of *agg* include the regional knowledge carrying capacity, ecological total factor productivity and regional pollution control performance [41–43]. The new energy industry needs financial support because it is a capital- and knowledge-intensive industry. At present, most countries in the world adopt the incentive policy of government subsidies, which theoretically will be advantageous for developing new energy industry. But excessive government subsidies will exacerbate the overcapacity of new energy enterprises. There are studies that show subsidies have crowded out the input of research and development of new energy enterprises in China and South Korea, which is not conducive to their technological innovation [44,45]. Therefore, The Party's 14th Five-Year Plan mentions

developing green finance. It aims to support green technology innovation and promote cleaner production. The current studies on green finance are dominated by China, the United Kingdom and Japan, mainly focusing on green finance and technological innovation, environmental performance and other aspects [46]. For example, Chinese scholars mostly study the implementation effect of policies. Among them, Liu et al. [47] explored the effect of Green Credit Guidelines on the innovation performance of heavily polluting enterprises by adopting the DID model. Xie and Liu [48] verified green credit's significant contribution to green economic growth. Research on green finance and the new energy industry mostly starts from the micro perspective, mainly studying their impact on corporate financing and corporate value [49,50]. At present, some literature has begun to investigate the relationship between financial support and industrial agglomeration; for example, Cao et al. [39] constructed financial support indicators to explore financial support's relationship with new energy industry agglomeration, and Zhao [51] found that *GF* would contribute to *agg*.

In summary, the existing literature mostly concentrates on the relationship between industrial agglomeration and energy efficiency [52], green finance and green innovation efficiency. However, the literature on the relationship between *agg* and *GI*, as well as the mechanism analysis of *GF*, is limited. Therefore, starting from China's provincial panel from 2012 to 2021, this paper first uses Super-SBM to estimate *GI* and then uses the location entropy to measure *agg*. Then, the GMM model is used to explore the effect of *agg* on *GI* and verify the regulatory mechanism of *GF*. There may be the following marginal contributions: first, further understanding the situation of *agg*, *GF* and *GI*, which is meaningful to enrich the existing studies; second, exploring the relation between *agg* and *GI* using the dynamic system GMM, which is greatly helpful for forming policies; third, taking *GF* as a regulating variable, which further enriches the literature on the mechanism of *agg* affecting *GI*.

## 3. Theoretical Analysis and Research Hypothesis

### 3.1. New Energy Industry Agglomeration and Green Innovation Efficiency

Green innovation efficiency should emphasize both "innovation" and "environmental attributes", so on account of the concept connotation of *GI*, this article will take "innovation" and "environment" as the starting point for analyzing the influence of *agg* on *GI*.

First of all, industrial agglomeration will increase the scale of enterprises [53] to further exert the scale effect. According to Marshall's external economy theory, industrial agglomeration will have significant positive external economic effects on enterprises in agglomeration areas through three mechanisms, namely intermediate input sharing, labor market formation and technology spillover, thus having a positive impact on green innovation efficiency [54]. Specifically, first, new energy industrial agglomeration can reduce costs through intermediate input sharing. This will further strengthen the cooperation between enterprises in the agglomeration areas, reduce production and research and development costs and reduce environmental pollution due to the intensive use of infrastructure, which will further improve *GI* [55]. Second, as a technology-intensive industry, the chain of the new energy industry presents a complex situation, which has extremely strict requirements for labor. Specialized labor and economies of scale will promote industrial development. In this context, various production factors will be gathered in the agglomeration area due to *agg*, and skilled labor will also be included. At this time, the growth of skilled labor and the creation of professional technology will promote *GI* [56]. Third, knowledge and technology spillover effect. The breadth, depth and frequency of intra-firm and inter-industry market exchanges will increase due to the geographical concentration of industries. As for the new energy industry, as a representative of the high-tech industry, it is bound to accumulate a lot of knowledge and experience in the process of development, and the industrial agglomeration will make this knowledge and experience transfer in space, resulting in a spillover effect. Therefore, other enterprises will make technological progress on the basis of accepting the above knowledge dissemination, which will be beneficial for industrial technological innovation in the region [41].

Secondly, while considering the positive role of the scale effect brought by *agg*, we should also pay attention to the inhibitory impact caused by the congestion effect [57]. When the agglomeration scale further expands and exceeds the supply of innovative factors and public goods, new energy enterprises will strengthen the competition for factor resources in order to maintain or expand market influence, which will lead to adverse competition and further have a negative impact on *GI* [58]. Moreover, the refinement of the division of labor in industrial agglomeration will lead to a production detour, which will further increase transaction costs and reduce the scale effect of the division of labor. In addition, from the perspective of knowledge spillover, excessive competition among enterprises will enhance their self-protection mechanism of innovation, so as to over-protect technology and inhibit the diffusion of knowledge and technology [59]. In short, the competition for production factors, the crowding of resources and facilities and the crowding of product markets will inhibit green innovation, which is more obvious in the mature period or even the decline period of industrial agglomeration.

Therefore, we make the following assumption:

**Hypothesis 1.** *agg can contribute to GI. However, the agglomeration scale will have an impact on its promoting effect; there may be a nonlinear relationship.*

### 3.2. New Energy Industry Agglomeration, Green Finance and Green Innovation Efficiency

*GF*, with the characteristic of integrating the concept of environmental protection and finance function, will have impacts on the *GI* of *agg* from the points of environmental regulation and capital allocation. In terms of environmental regulation, high green financial regions symbolize that there will be stricter environmental regulation and more foreign investment. Advanced technology level and environmental protection experience will be introduced with the development of foreign investment, which will enable the role of knowledge to be strengthened, thus improving *GI* [60]. In addition, the "innovation compensation" effect which resulted from environmental regulation will enhance new energy enterprises developing within the aggregation area, so that *GI* will become higher. From the aspect of financial functions, high-polluting enterprises with heavy assets can extremely easily receive funds from financial institutions because financial institutions have the characteristic of agglomeration and are profit-driven [61]. So, the situation is extremely difficult for green innovation industries with high costs and high risks, which leads to significant resource misallocation. Green finance can alleviate resource misallocation. Low green finance indicates that there are less funds flowing into the new energy industry. In this case, enterprises will maintain their current operating scale, which will lead to a reduction in R&D investment. Therefore, the innovation activities of enterprises will be influenced, which is not conducive to the external economies of scale [62]. If *GF* is high, it is easier for new energy enterprises to obtain financial support. The free flow of capital enables enterprises to conduct R&D activities with confidence, which further promotes external economies of scale and further strengthens *GI*. When *GF* is too high, the aggregation area will attract more new energy enterprises to enter the region, in the case of certain element resources, which will intensify enterprises' competition and weaken the influence of *agg* on *GI*. Therefore, according to the above analysis, assumptions are put forward in turn.

**Hypothesis 2.** *Green finance will promote the effect of new energy industry agglomeration on green innovation efficiency.*

**Hypothesis 3.** *With the improvement of GF, the positive effect of agg on GI will gradually weaken.*

## 4. Methodology

### 4.1. Estimation Methods

A dynamic panel regression model is a regression model that includes the lagged term of the explained variable in the panel data. Among dynamic panel regression models, the GMM model is widely used. This model can solve the endogeneity problem between variables, the correlation of explanatory variables and the error caused by time. Although two-stage least squares (2SLS) can also solve the above problems, GMM assumptions are more relaxed than those of 2SLS. Therefore, the problems of heteroscedasticity and autocorrelation will be solved by the GMM model [63]. Meanwhile, the research data are short panel data, with the characteristic of strong balance, and are suitable for the system GMM approach. Due to the intertemporal continuity of *GI*, a dynamic panel regression model is constructed by introducing the one-period-lagged variable of green innovation efficiency to probe into the impact of *agg* on *GI*. The equations are as follows:

$$GI_{i,t} = \omega_1 GI_{i,t-1} + \alpha_1 agg_{i,t} + \alpha_n X_{i,t} + \mu_i + \varepsilon_{i,t} \tag{1}$$

$$GI_{i,t} = \omega_2 GI_{i,t-1} + \alpha_2 mar_{i,t} + \alpha_n X_{i,t} + \mu_i + \varepsilon_{i,t} \tag{2}$$

$$GI_{i,t} = \omega_3 GI_{i,t-1} + \alpha_3 jac_{i,t} + \alpha_n X_{i,t} + \mu_i + \varepsilon_{i,t} \tag{3}$$

where $GI_{i,t}$ represents the green innovation efficiency of region $i$ in period $t$; $agg_{i,t}$ reflects the agglomeration level of the new energy industry in region $i$ in period $t$; $mar_{i,t}$ and $jac_{i,t}$ refer to the professional agglomeration level and diversified agglomeration level of the new energy industry in region $i$ in period $t$; $X_{i,t}$ describes the control variables. In addition, $\mu_i$ is the individual fixation effect; $\varepsilon_{i,t}$ is the random perturbation term.

In order to further explore the moderating effect of *GF*, this article adds the interaction term of *GF* and *agg*, specialization (*mar*) and diversified agglomeration (*jac*) of the new energy industry into the model, as shown in Equations (4)–(6):

$$GI_{it} = \omega_4 GI_{it-1} + \beta_1 agg_{it} + \beta_2 GF_{it} + \beta_3 GF_{it} \times agg_{it} + \beta_n X_{it} + \lambda_i + \varepsilon_{it} \tag{4}$$

$$GI_{it} = \omega_5 GI_{it-1} + \beta_{11} mar_{it} + \beta_{21} GF_{it} + \beta_{31} GF_{it} \times mar_{it} + \beta_n X_{it} + \lambda_i + \varepsilon_{it} \tag{5}$$

$$GI_{it} = \omega_6 GI_{it-1} + \beta_{12} jac_{it} + \beta_{22} GF_{it} + \beta_{32} GF_{it} \times jac_{it} + \beta_n X_{it} + \lambda_i + \varepsilon_{it} \tag{6}$$

In this moderating model, we will focus on three coefficients: $\beta_3$, $\beta_{31}$ and $\beta_{32}$.

Finally, the threshold regression model is used to explore whether there is a threshold value between *agg* and *GI*. Existing scholars mostly use the Hansen model to conduct static panel regression, but the endogeneity between variables will cause estimation bias at this time. So, on the basis of the baseline regression model, adopting the system GMM estimation method, we make *agg* and *GF* threshold variables to explore the effect of *agg* on *GI*. The model is as follows:

$$GI_{it} = \omega_7 GI_{it-1} + \gamma_{11} agg_{it} I(threshold_{it} \leq \eta_1) + \gamma_{11} agg_{it} I(threshold_{it} > \eta_1) + \gamma_{n1} X_{it} + \mu_i + \varepsilon_{it} \tag{7}$$

$$\begin{aligned} GI_{it} = \omega_8 GI_{it-1} &+ \gamma_{21} agg_{it} I(threshold_{it} \leq \eta_1) + \gamma_{22} agg_{it} I(\eta_1 < threshold_{it} \leq \eta_2) \\ &+ \gamma_{23} agg_{it} I(threshold_{it} > \eta_2) + \gamma_{n1} X_{it} + \mu_i + \varepsilon_{it} \end{aligned} \tag{8}$$

Formula (7) is expressed as the single-threshold model of *agg* on *GI*; Equation (8) represents the two-threshold model. Here, threshold represents the threshold variable, namely *agg* and *GF*. $I(\cdot)$ is the indicative function, for example, in Equation (7), if $threshold_{it} \leq \eta_1$ is true, $I(threshold_{it} \leq \eta_1)$ is 1; if $threshold_{it} \leq \eta_1$ is false, $I(threshold_{it} \leq \eta_1)$ is 0. $\eta_1$ and $\eta_2$ represent the threshold values.

### 4.2. Variable Description

#### 4.2.1. Explained Variable

The explained variable is green innovation efficiency (*GI*). Green innovation takes into account innovation output, economic growth and environmental benefits, so the Super-

SBM model considering unexpected output is used to calculate *GI*. On the basis of the work of Zhang et al. [64], the evaluation index system for *GI* is established. The variables are selected as follows:

(1) Input factors. Input factors mainly include capital, labor and energy input. Capital input is measured by the government's scientific expenditure, labor input is calculated by the number of employees engaged in scientific research and technical services and energy input is mainly calculated by considering two terms: one is the total amount of regional water supply, and the other is the electricity consumption of the whole society.

(2) Expected output. Achieving innovative growth and green development is the ultimate goal of green innovation. Innovative growth also includes innovation output and economic growth. Therefore, the expected output in this paper is determined by regional GDP per capita and the total number of patents granted. Regional GDP per capita here is deflated in 2012.

(3) Undesirable output. Considering the green development purpose of green innovation, environmental pollution must be minimized while the expected output is maximized. On the basis of considering data continuity, three industrial wastes are selected to represent the undesirable benefits.

### 4.2.2. Explanatory Variables

(1) Agglomeration level of new energy industry (*agg*)

At present, the method of location entropy is the most widely used method in academia, and a large number of studies have verified the stability of this method. Based on the work of Guo Liwei et al. [38] and Yan et al. [41], the location quotient is used to calculate *agg*. The specific formula is as follows:

$$agg_{it} = LQ_{it} = \frac{E_{it}/Y_{it}}{E_t/Y_t} \tag{9}$$

In Equation (9), $E_{it}$ means the new energy output value of region $i$ in period $t$, and $E_t$ reflects the national value in the same period; $Y_{it}$ is the industrial output value of region $i$ in period $t$, and $Y_t$ reflects the whole country in the same period. The larger the value is, the more concentrated the industry is in the region. In order to further explore the influence of different agglomeration modes on *GI*, this article further subdivides *agg* into *mar* and *jac*.

(2) Specialization agglomeration index (*mar*)

The Krugman specialization index is adopted to represent the level of industrial specialization agglomeration, and its formula is as follows [65]:

$$mar = \sum_{i=1}^{I} \left| \frac{R_{j,i}}{R_i} - \frac{R_j}{R} \right| \tag{10}$$

where $j$ represents industry; $i$ denotes province; $R$ represents the industrial output value. A higher ksl index indicates that there is a higher level of industrial professional aggregation in the region.

(3) Diversified agglomeration index (*jac*)

On the basis of referring to Xie Guo et al. [66], the index is measured by the inverse div index of the HHI index, and the formula is as follows:

$$jac = \frac{1}{\sum_j (R_{ji}/R_i)^2} \tag{11}$$

The meaning of the letters is the same as above.

(4) Green finance (*GF*)

Green finance, as the extension of environmental regulation policy, symbolizes the financial support for green industries. Based on the existing literature, green credit, green securities, green insurance and green investment are included in the green finance index system [67,68]. Then, *GF* is calculated by the entropy method. Table 1 shows the specific indicators.

**Table 1.** Indicator system for *GF*.

| | First-Level Indicator | Second-Level Indicator | Third-Level Indicator |
|---|---|---|---|
| **Green finance development level** | Green credit | Interest ratio of six high-energy-consumption industries | High-energy-consumption industry interest/industrial industry interest |
| | Green securities | Proportion of market value of environmental protection enterprises | Market value of environmental protection enterprises/total market value of A shares |
| | | Proportion of market value of energy-intensive industries | Market value of high energy consuming industry/total market value of A shares |
| | Green insurance | Depth of agricultural insurance | Agricultural insurance income/total agricultural output value |
| | | Agricultural insurance loss ratio | Agricultural insurance expenditure/agricultural insurance income |
| | Green investment | Proportion of public expenditure on energy conservation and environmental protection | Fiscal expenditure on energy conservation and environmental protection/GDP |
| | | Proportion of investment in environmental pollution control | Investment in pollution control/GDP |

4.2.3. Control Variables

This article chooses the following variables as control variables: (1) the level of transportation infrastructure (*trans*) [69], which is measured by the ratio of the total mileage of expressways to the total population; (2) the degree of government intervention (*gov*) [69], which is measured as the share of fiscal expenditure of government spending minus expenditures on science and technology, education, culture and health as a percentage of GDP; (3) ownership structure (*own*), which is the proportion of the main business income of state-owned and state-controlled enterprises; (4) environmental regulation [70] (*envir*), which is calculated by the ratio of environmental pollution emissions to industrial output value.

*4.3. Data Source*

Since the "Twelfth Five-Year Plan for Renewable Energy Development" was released in 2012, the year 2012 is selected as the start year. Considering the availability of data, the year 2021 is determined as the last year of the research. In addition, energy data are lacking in Hong Kong, Macau, Taiwan and Tibet, so these areas are excluded from the sample. In summary, this paper focuses on 30 regions in China from 2012 to 2021. Currently, the specific statistics on the new energy industry have not been compiled by specific departments, so when the text calculates *agg*, it is important to review China's concept stock sector. The sector covers charging piles, energy storage, wind energy, nuclear power, lithium batteries, green power, hydrogen energy, fuel cells, biomass energy, solar energy, new energy vehicles and smart grids. Then, according to the main business income, 595 new energy enterprises are selected. Considering the industry distribution of 595 new energy enterprises, we select electrical machinery and equipment manufacturing; computer, communication and other electronic equipment manufacturing; automobile manufacturing; general equipment manufacturing; and special equipment manufacturing for output value substitution. All data resources are found in the China Industrial Statistical Yearbook, provincial statistical bureaus and EPS statistical databases. This paper deals with missing

values and related data through interpolation and logarithmic processing. The descriptive statistics of the relevant variables are displayed in Table 2.

**Table 2.** The descriptive statistics of each variable.

| Variable | Obs | Mean | Std. Dev. | Min | Max |
|---|---|---|---|---|---|
| GI | 300 | 0.582 | 0.394 | 0.158 | 1.585 |
| agg | 300 | 0.785 | 0.496 | 0.0176 | 2.037 |
| mar | 300 | 0.176 | 0.0846 | 0.0486 | 0.674 |
| jac | 300 | 4.729 | 1.636 | 1.241 | 12.30 |
| GF | 300 | 0.214 | 0.0675 | 0.113 | 0.502 |
| trans | 300 | 1.200 | 0.858 | 0.333 | 5.897 |
| gov | 300 | 0.191 | 0.0904 | 0.0672 | 0.601 |
| own | 300 | 0.351 | 0.172 | 0.0600 | 0.810 |
| envir | 300 | 0.723 | 1.114 | 0.00676 | 6.595 |

## 5. Empirical Results and Discussion

### 5.1. Results of Spatiotemporal Differentiation

5.1.1. The Temporal and Spatial Changes in New Energy Industry Agglomeration

The results for *agg* in the provinces of China from 2012 to 2021 are reflected in Figure 1. In this picture, if the color is brown, the *agg* is superior. In this picture, provinces like Beijing, Shanghai, Jiangsu, Guangdong, Jilin and Chongqing possess larger LQ values. This indicates that agglomeration has formed in these areas. The reasons for agglomeration in these regions are more developed economies and foreign trade. Meanwhile, the locations of these provinces are better than others, with a relatively complete industrial base, including the Yangtze River Delta, Pearl River Delta, and Beijing–Tianjin–Hebei region. This is consistent with the work of Wang et al. [33]. However, the LQ values of Yunnan, Gansu, Qinghai, Ningxia, Xinjiang and Inner Mongolia are much smaller, which indicates that *agg* in these regions is low. In addition, the LQ of different regions can change over time. In the picture, the LQ of the eastern regions such as Beijing and Shanghai has begun to show a downward trend, while that of the central provinces such as Jiangxi, Hubei and Hunan has further improved, which may be closely related to national policies. In short, in general, *agg* presents a distribution of "the highest in the eastern region, followed by the central region, and the lowest in the western region".

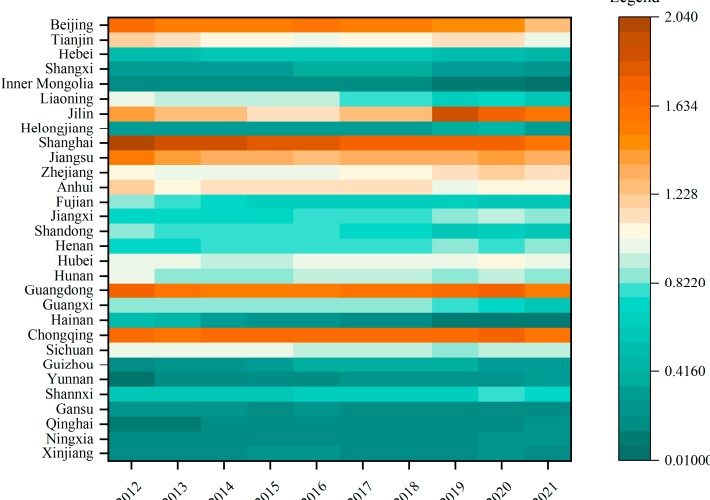

**Figure 1.** LQ of the new energy industry in China's provinces from 2012 to 2021.

5.1.2. Spatiotemporal Analysis of Green Finance Development Level

Figure 2 shows *GF* in Chinese provinces from 2012 to 2021. According to Figure 2, Beijing, Shanghai, Qinghai, Ningxia, Xinjiang and other places have good development. Beijing and Shanghai, as the representatives of developed provinces in eastern China, have obvious advantages in developing green finance. For Xinjiang, Ningxia and Xinjiang, *GF* was cut off in 2014, and these provinces mainly benefited from the promulgation of national policies and the implementation of national strategies, such as the Western Development strategy and the great initiative of "The Belt and Road", which made green finance effectively developed. This result is in agreement with the research of some scholars [18]. In addition, compared with 2014 and the situation in central provinces such as Jiangxi, Hunan and Henan in 2021, *GF* in eastern regions such as Shandong and Guangdong has improved, mainly due to the Guidance on Building a Green Financial System issued by the state in 2017. On the whole, there are obvious diversities in *GF* in various provinces, and the features are unbalanced and inadequate.

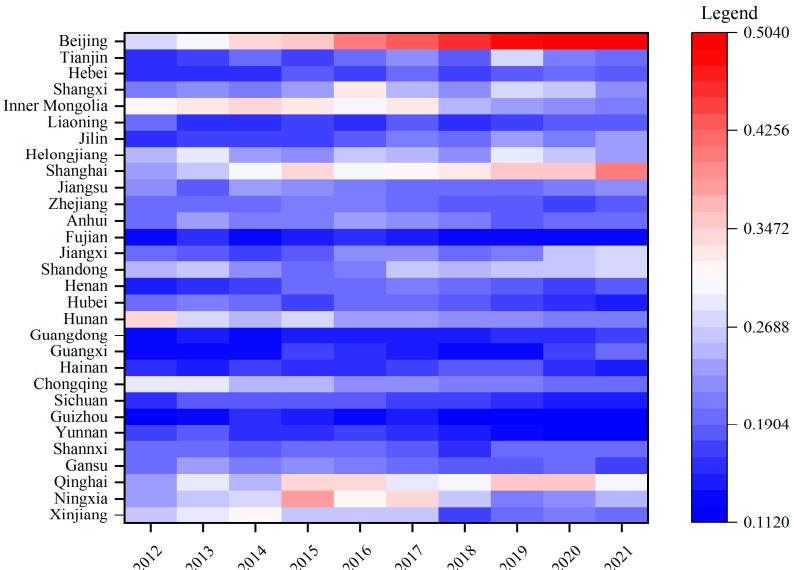

**Figure 2.** The development level of green finance in China's provinces from 2012 to 2021.

*5.2. Results of Statistical Tests*

5.2.1. Results of Baseline Regression

Based on the work of Hou et al. [71], aiming to alleviate the problems of endogeneity, heteroscedasticity and autocorrelation, the text chooses the system GMM method. The results are displayed in Table 3.

In accordance with Table 3, the *p*-values of AR(1) and AR(2) tests pass the autocorrelation test, and the Sargan test also passes the correlation test. The results revealed that the system GMM estimation result is valid.

In Table 3 Column (1), without the addition of control variables, the influence coefficient of new energy industry agglomeration is 0.1600 ($p < 0.01$), which indicates that *agg* has a positive effect on *GI*. In Column (5), with the addition of control variables, the coefficient becomes 0.1543 ($p < 0.01$), which indicates that the positive impact still stands. This result verifies the first half of Hypothesis 1, which has been confirmed by other researchers [69]. Column (2) and Column (4) are used to study the effect of *mar* on *GI*. When Column (4) is compared to Column (2), with the addition of control variables, the influence coefficient becomes 0.6010 ($p < 0.01$), which indicates that *mar* can improve *GI*. The influence coefficients of Model (3) and Model (6) on green innovation efficiency are −0.0418 and −0.0409 ($p < 0.01$), which verifies that the diversified agglomeration level of new energy industries will have an inhibitory effect on green innovation efficiency. This may be related to the low correlation between diversified industries in a region, which results in a circular learning

system with a complete network structure being unformed within the industry. Then, the scale and technological progress effects generated by *jac* do not appear, thus inhibiting the improvement of *GI* [72]. Moreover, on the basis of Table 3, *mar* is the key way that *agg* has a positive influence on *GI*.

**Table 3.** Baseline regression results.

| Variable | GI | | | | | |
|---|---|---|---|---|---|---|
| | **(1)** | **(2)** | **(3)** | **(4)** | **(5)** | **(6)** |
| *L.GI* | 0.7384 *** 0.0107 | 0.6962 *** 0.0070 | 0.7516 *** 0.0200 | 0.7847 *** 0.2301 | 0.8412 *** 0.0207 | 0.7736 *** 0.0370 |
| *agg* | 0.1600 *** 0.0058 | | | 0.1545 *** 0.0449 | | |
| *mar* | | 0.7873 *** 0.0132 | | | 0.6010 *** 0.1203 | |
| *jac* | | | −0.0418 *** 0.0034 | | | −0.0409 *** 0.0049 |
| *trans* | | | | 0.0449 *** 0.0046 | 0.0141 *** 0.0038 | 0.0325 *** 0.0061 |
| *gov* | | | | 0.4170 *** 0.0736 | −0.0226 0.0438 | 0.4920 *** 0.1209 |
| *own* | | | | 0.0139 0.0626 | −0.1941 *** 0.0261 | −0.0294 0.0736 |
| *envir* | | | | −0.0256 *** 0.0037 | −0.0050 0.0045 | −0.0275 *** 0.0102 |
| _cons | | | | −0.1099 ** 0.0509 | 0.0532 ** 0.0215 | 0.2276 *** 0.0622 |
| AR(1) | 0.0204 | 0.0191 | 0.0174 | 0.0189 | 0.0153 | 0.0196 |
| AR(2) | 0.4167 | 0.3827 | 0.3591 | 0.4169 | 0.3918 | 0.3655 |
| Sargan test | 0.9877 | 0.9953 | 0.9965 | 1.0000 | 1.0000 | 1.0000 |
| observation | 270 | 270 | 270 | 270 | 270 | 270 |

Legend: * $p < 0.1$; ** $p < 0.05$; *** $p < 0.01$.

In terms of control variables, the influence coefficients of transport infrastructure level and the degree of government intervention are remarkably positive, while the influence coefficient of environmental regulation is significantly negative. The results suggest that transport infrastructure and government intervention can contribute to green innovation efficiency while environmental regulation can inhibit green innovation efficiency. The reasons are as follows: Firstly, transport infrastructure can promote communication among enterprises and reduce loss of information. Secondly, government intervention can make some policies to contribute to *GI*. Thirdly, environmental regulation will increase the cost of enterprises' environmental utilization and squeeze out profits, thus affecting R&D innovation investment.

In summary, the benchmark regression results verify the first half of Hypothesis 1.

### 5.2.2. Robustness Test

In order to verify the robustness of the results, this paper chooses to change the model to re-estimate the regression. Table 4 shows the results. The influence coefficients of *agg* and mars are still positive, while the influence coefficient of *jac* is still negative. The significance still passes. So, the empirical regression results are robust and reliable.

**Table 4.** Robustness test.

| Variable | GI | | | | | |
|---|---|---|---|---|---|---|
| | **(7)** | **(8)** | **(9)** | **(10)** | **(11)** | **(12)** |
| *L.GI* | 0.7818 *** 0.0037 | 0.4638 *** 0.0067 | 0.6036 *** 0.0152 | 0.5983 *** 0.0324 | 0.3747 *** 0.0288 | 0.4914 *** 0.0213 |
| *agg* | 0.4392 *** 0.0178 | | | 0.1672 * 0.0905 | | |
| *mar* | | 1.7142 *** 0.0536 | | | 1.7621 *** 0.3004 | |
| *jac* | | | −0.0953 *** 0.0114 | | | −0.0538 *** 0.0133 |
| *trans* | | | | 0.0928 *** 0.0108 | 0.0659 *** 0.0086 | 0.0599 *** 0.0146 |
| *gov* | | | | 1.2286 *** 0.0859 | 0.3159 *** 0.0558 | 1.1718 *** 0.1498 |
| *own* | | | | 0.4411 *** 0.0479 | −0.0383 0.0475 | 0.5118 *** 0.0440 |
| *envir* | | | | −0.1136 *** 0.0116 | −0.0581 *** 0.0089 | −0.0877 *** 0.0171 |
| _cons | −0.2098 *** 0.0118 | 0.01604 ** 0.0065 | 0.6820 *** 0.0524 | −0.3019 *** 0.0680 | −0.0311 0.0467 | 0.1498 * 0.0842 |
| AR(1) | 0.0221 | 0.0249 | 0.0248 | 0.0173 | 0.0302 | 0.0239 |
| AR(2) | 0.5729 | 0.3942 | 0.3807 | 0.3610 | 0.3782 | 0.2893 |
| Sargan test | 0.8457 | 0.7552 | 0.8043 | 1.0000 | 1.0000 | 1.0000 |
| observation | 240 | 240 | 240 | 240 | 240 | 240 |

Legend: * $p < 0.1$; ** $p < 0.05$; *** $p < 0.01$.

### 5.2.3. Results of Moderating Effect Regression

Table 5 reports the results of *GF* as a regulating variable on *agg* and *GI*. In Column (16), we can see that the influence coefficient of *agg* is 0.1806 and passes the significance test. The first half of Hypothesis 1 is further confirmed. In addition, the coefficient of the interaction term is 0.5850 and also passes the significance test. According to this, it can be concluded that *GF* will promote the effect of *agg* on *GI*, thus verifying Hypothesis 2 of the theoretical analysis. This result is consistent with other scholars' studies [39].

In addition, the moderating effect of *GF* on *GI* of different types of industrial agglomeration is also discussed in this paper. Compared to Column (14), Column (17) adds control variables, exploring the moderating effect of *GF* on the *GI* of *mar*. In Column (17), the coefficient of the interaction term is significantly positive. Column (18) compared to column (15), with the addition of control variables, reveals the moderating effect of *GF* on the *GI* of *jac*. On the contrary, the coefficient is remarkably negative. The results suggest that *GF* has a moderating effect on the *GI* of *agg* mainly by affecting the specialized agglomeration of new energy. This is because green finance can promote foreign investment [61], thus bringing more advanced technological achievements, which can better enable *mar* to play its external economy role and thus promote the improvement of the *GI* of *agg*.

According to the above analysis, the moderating effect regression verifies Hypothesis 2, that is, *GF* will promote the positive influence of *agg* on *GI*.

**Table 5.** The results of moderating effects.

| Variable | GI | | | | | |
|---|---|---|---|---|---|---|
| | (13) | (14) | (15) | (16) | (17) | (18) |
| L.GI | 0.7362 *** 0.0131 | 0.7074 *** 0.0051 | 0.6351 *** 0.0178 | 0.7548 *** 0.0353 | 0.8289 *** 0.2903 | 0.8472 *** 0.0316 |
| agg | 0.0933 *** 0.0078 | | | 0.1806 *** 0.0435 | | |
| GF | 0.5575 *** 0.1245 | 0.6692 *** 0.0383 | 0.9405 *** 0.0513 | 0.5579 *** 0.1369 | 0.4708 *** 0.1112 | 0.5312 *** 0.1514 |
| GF × agg | 0.1708 0.2331 | | | 0.5850 *** 0.1881 | | |
| mar | | 0.4057 *** 0.0427 | | | 0.4000 * 0.2406 | |
| GF × mar | | 1.6397 *** 0.7096 | | | 5.3587 * 3.1083 | |
| jac | | | −0.0230 *** 0.0049 | | | −0.0197 *** 0.0065 |
| GF × jac | | | −0.2996 *** 0.0218 | | | −0.4334 *** 0.0870 |
| trans | | | | 0.0487 *** 0.0059 | 0.02511 *** 0.0038 | 0.0311 *** 0.0060 |
| gov | | | | 0.3766 *** 0.1125 | −0.0506 0.0915 | 0.5411 *** 0.1331 |
| own | | | | 0.1692 0.1055 | −0.1377 * 0.0805 | −0.2078 ** 0.0856 |
| envir | | | | −0.0455 *** 0.0084 | −0.0158 ** 0.0074 | −0.0322 *** 0.0061 |
| _cons | 0.1618 *** 0.0101 | 0.1827 *** 0.0078 | 0.2199 *** 0.0111 | −0.0051 0.0359 | 0.14488 *** 0.0208 | 0.0613 ** 0.0303 |
| AR(1) | 0.0199 | 0.0184 | 0.0188 | 0.0227 | 0.0187 | 0.0145 |
| AR(2) | 0.3799 | 0.3607 | 0.3227 | 0.3700 | 0.3466 | 0.2825 |
| Sargan test | 1.0000 | 1.0000 | 1.0000 | 1.0000 | 1.0000 | 1.0000 |
| observation | 270 | 270 | 270 | 270 | 270 | 270 |

Legend: * $p < 0.1$; ** $p < 0.05$; *** $p < 0.01$.

5.2.4. Results of Threshold Regression

Aiming to further study the complex heterogeneity mechanism of the effect of *agg* and *GF* on *GI*, *agg* and *GF* are taken as threshold variables to test the following three hypotheses: (1) $H_0^I$: there is no threshold, $H_1^I$: there is a threshold; (2) $H_0^{II}$: there is a single threshold, $H_1^{II}$: there are two thresholds; (3) $H_0^{III}$: there are only two thresholds, $H_1^{III}$: there are three thresholds. Table 6 shows the results of the threshold test. The significance test value is 0.3533, which rejects the double-threshold test of *agg*, showing that only one threshold value exists. As for *GF* as the threshold variable, the test finds that there are double thresholds. The threshold value of *agg* is 1.3763, and the threshold values of *GF* are 0.1439 and 0.2134 (see Table 7).

**Table 6.** Test results of threshold effects.

| Threshold Variable | Threshold Effect | F Value | *p* Value | 10% | 5% | 1% |
|---|---|---|---|---|---|---|
| *agg* | Single threshold | 31.67 | 0.0667 | 27.7935 | 36.5676 | 58.4722 |
| | Double threshold | 16.9 | 0.3533 | 27.7707 | 36.6151 | 51.6785 |
| *GF* | Single threshold | 33.38 | 0.0333 | 22.2718 | 28.5545 | 58.0677 |
| | Double threshold | 23.86 | 0.0833 | 21.8228 | 28.3012 | 40.9749 |
| | Triple threshold | 11.57 | 0.7233 | 32.3263 | 40.2469 | 49.7172 |

**Table 7.** Threshold values and confidence intervals.

| Threshold Variable | Test | Threshold Estimates | 0.95 Confidence Interval |
|---|---|---|---|
| *agg* | Single-threshold value | 1.3763 | [1.3666, 1.3915] |
| *GF* | Single-threshold value | 0.1439 | [0.1433, 0.1455] |
| | Double-threshold value | 0.2134 | [0.2097, 0.2144] |

Table 8 reports the regression results of *agg* and *GI* when *agg* and *GF* are taken as the threshold variables. The result in Column (19), with *agg* as the threshold variable, reveals that *agg* can further contribute to *GI*, but the influence coefficient is different. This verifies the second half of Hypothesis 1. Concretely, the influence coefficient is 0.0407 ($p < 0.01$), with the new energy industry agglomeration being less than 1.3763. But when *agg* is greater than 1.3763, the influence coefficient is 0.2300 ($p < 0.01$). This is consistent with the current research of most scholars [24,25]. But it is not consistent with Li's research, which suggests that excessive agglomeration will obscure the *GI* of *agg* [69]. The main reason may be that *agg* in China is not high as a whole.

**Table 8.** The result of threshold model.

| Variable | GI | |
|---|---|---|
| | (19) | (20) |
| *L.GI* | 0.580 *** −0.00716 | 0.516 *** −0.0274 |
| *agg* (*agg* ≤ 1.3763) | 0.0407 *** −0.0138 | |
| *agg* (*agg* > 1.3763) | 0.2300 *** −0.0111 | |
| *agg* (*GF* ≤ 0.1439) | | −0.340 ** −0.151 |
| *agg* (0.1439 < *GF* ≤ 0.2134) | | 0.388 *** −0.0583 |
| *agg* (*GF* > 0.2134) | | 0.260 *** −0.0176 |
| *trans* | 0.0544 *** −0.00806 | 0.0588 *** −0.0139 |
| *gov* | 0.325 *** −0.0798 | 0.551 *** −0.0903 |
| *own* | 0.196 *** −0.0363 | 0.287 *** −0.0631 |
| *envir* | −0.0215 *** −0.00786 | −0.0431 *** −0.00877 |

**Table 8.** *Cont.*

| Variable | GI | |
|---|---|---|
| | **(19)** | **(20)** |
| Constant | −0.0132<br>−0.0228 | −0.150 ***<br>−0.0502 |
| AR(1) | 0.0388 | 0.0298 |
| AR(2) | 0.4215 | 0.4985 |
| Sargan test | 0.9962 | 0.9933 |
| Observations | 270 | 270 |

Legend: * $p < 0.1$; ** $p < 0.05$; *** $p < 0.01$.

Column (20) displays the results obtained when *GF* is taken as the threshold variable. These results reveal that the promotion effect of *agg* on *GI* is gradually weakened with the double-threshold effect of green finance. Hypothesis 3 is confirmed. From the specific numerical point of view, when the level of green finance is lower than 0.1439, the influence coefficient is negative, which indicates that under the condition of a limited local financial level, the scale effect will be weakened. When *GF* crosses a single-threshold value ($0.1439 < GF \leq 0.2134$), the influence coefficient is positive, and *agg* has the strongest driving effect on *GI*, with a coefficient of 0.388. At this time, enterprises in the area have sufficient funds, allowing them to better carry out technological innovation and share technical knowledge, and the labor division and other markets are more active. In addition, green finance also makes high energy consumption and high pollution move out of the area, thus improving *GI* in the area [51]. But with the continuous improvement of *GF*, the region becomes more suitable for the development of new energy enterprises, which will lead to a large number of enterprises entering the region. The phenomenon will cause intensified competition and a scramble for resources, which will hinder the promotion of *GI*.

The second half of Hypothesis 1, Hypothesis 2 and Hypothesis 3 are verified.

## 6. Conclusions and Implication

### 6.1. Conclusions

With the implementation of China's innovation-driven high-quality economic development strategy, green and innovation have become the key factors of economic development. Therefore, developing green industry and improving green innovation have attracted wide attention and are of great significance to the sustainable development of China's economy. Therefore, starting from China's provincial panel from 2012 to 2021, this article first uses Super-SBM to measure *GI* and then uses location entropy to calculate *agg*. Then, the GMM model is used to explore the impact of *agg* on *GI* and verify the regulatory mechanism of green finance. The text enriches the existing literature on the development of the new energy industry, green innovation efficiency and green finance. The conclusions are as follows:

(1) At present, *agg* presents a distribution of "the highest in the eastern region, followed by the central region, and the lowest in the western region". (2) *GF* in Beijing, Shanghai, Ningxia and other places is higher than that in other places, which is closely related to national policies. This is consistent with the research of Chinese domestic scholars [73]. (3) *agg* can promote *GI*; when the threshold value (1.3763) is exceeded, the promotion effect is more significant. Moreover, we find that the *agg* mainly promotes *GI* through *mar*. (4) *GF* has a positive moderating effect on *agg* and *GI*, but when *GF* exceeds 0.2134, the positive moderating effect weakens.

### 6.2. Policy Suggestion

Based on the research results, policy suggestions are proposed:

(1) The state should carry on vigorously promoting the implementation of green finance policies. As the development of green finance in China is dominated by green credit, the balance of green credit grew to CNY 15.9 trillion by the end of 2021, accounting for about 95 percent of green finance [74]. So, green credit deserves attention, and its implementation status should be supervised to ensure that green enterprises can receive funds to conduct technology innovation, which can provide a better capital environment for *agg*. The measures will promote the efficiency of green innovation in the region.

(2) New energy industry agglomeration should be guided by local governments, which should adopt appropriate policies that are determined by local resource conditions and economic levels. Currently, the industry is still in the stage of rapid development, but a gathering of new energy has not been formed, and it is mostly gathered in the eastern provinces with high economic development levels. Other provinces should give full play to their own unique advantages, such as talents and foreign investment; attract new energy enterprises to gather; and encourage enterprises to cooperate with schools and enterprises to promote more innovative output.

(3) Resources should be shared among enterprises in the agglomeration area, and large enterprises should achieve their leading role and help other enterprises in the agglomeration area in terms of talents and technology.

### 6.3. Limitations and Future Direction

Although this research provides some new ideas for the high-quality development of China's economy, it has limitations for future research. Firstly, due to the limitation of data availability, we only used provincial panel data, which will be improved with the development of the new energy industry. Secondly, this article studies the relationship between *agg* and *GI*, as well as the moderating impact of green finance. The spatial spillover effect of *agg* has not been studied, and its spatial spillover effect will be further discussed in the future. Finally, in the era of global commitment to the development of the new energy industry, the risks of its industrial chain and supply chain, including the non-ferrous metal industry, have also attracted great attention. Therefore, it is of great theoretical and practical value to study the effect of the co-agglomeration of the non-ferrous metal industry and the new energy industry in the future.

**Author Contributions:** Conceptualization, Y.W. and J.S.; methodology, Y.W.; software, J.S.; validation, Y.W.; formal analysis, Y.W.; investigation, J.S.; resources, J.S.; data curation, J.S.; writing—original draft preparation, J.S.; writing—review and editing, Y.W.; visualization, Y.W.; supervision, Y.W.; project administration, Y.W.; funding acquisition, Y.W. All authors have read and agreed to the published version of the manuscript.

**Funding:** The research was funded by Chinese National Funding of Social Sciences, grant number 20XGL016.

**Institutional Review Board Statement:** Not applicable.

**Informed Consent Statement:** Not applicable.

**Data Availability Statement:** The data that support the findings of this study are available from the corresponding author upon reasonable request.

**Conflicts of Interest:** The authors declare no conflicts of interest.

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
