# Peer review of "Impact of New Energy Industry Agglomeration on Green Innovation Efficiency—Based on the Regulative Effect of Green Finance"

_sustainability, doi:10.3390/su16083311_

Round 1

Reviewer 1 Report

Comments and Suggestions for Authors

“This paper measures the efficiency of green innovation, the agglomeration of new energy industry and the development level of green finance, and studies the impact of the agglomeration of new energy industry on the efficiency of green innovation and the mechanism analysis of green finance”.

Data: provincial Chinese panel data from 2012 to 2021

The manuscript must be improvement in some issues:
-    the main objective of the article must be explicitly in the abstract.
-    Acronyms must be introduced the first time (e.g.: SBM; GMM, DEA, SFA….).
-    The references are not correct (e.g.: page 42, 78, 81, 83…).
-    Text without bibliographic references (e.g.: lines 23-32, lines 43-50, lines 155-191….).
-    If the chosen of the variables was based on other studies, should be mentioned.
-    Equations 4, 5, 6 – not explain the GF – acronym introduction.
-    Explain better “I(·) is the indicative function”.
-    Tables needs a better representation, especially the table 1.
-    If the results are supported or not with the results of other authors, should be mentioned.
-    Explain better the results and the validation of hypotheses.
-    Conclusion – explain better the contributions of the study.

Reviewer 2 Report

Comments and Suggestions for Authors

The paper is interesting, the topic is challenging, promising inspiring openings, useful both for theory and practice regarding the impact of new energy industry agglomeration on green innovation efficiency.

The paper has merits, it is well organized, using a solid scientific and logical tool. Methodology and approaches are interesting, systematic and comprehensive.

However, I would have some considerations and suggestions for improving the quality of the article.

Content

The abstract, although precise, starts suddenly, specifying the method and the main results, without briefly describing the context, motivation and importance of the present research. A comprehensive approach would greatly help potential readers interested in this topic to have a suggestive image of the content and, consequently, to continue (or not) reading the article.

The literature review is well constructed, with relevant information, substantiating the actual research approach. We recommend, however, the diversification of analysed contributions. Although we acknowledge the research topic are focused on the exceptional situation of China - we consider there are other contributions from the international literature that can help build a complete theoretical picture of the main research directions on this topic.

We would recommend that, in the final part of the Results section, the authors do a systematic review of the issued hypotheses, indicating the final outcomes of their testing - accepted, partially accepted, rejected etc.

Also, we recommend a short discussion on the main limitation of the present research, and how do the authors intend to solve them in future papers. An explanatory sentence in this context, accompanied by the issue of future research topics (as a kind of invitation to academic debate) can help a lot at the end of the article's conclusion.

Minor formal issues –

-          Double hyphen in the title,

-          Citation style (numerical indications are in exponent form)

-          Most of the citations in the text include the first and last names of the authors (if they can be identified in this form) although, usually, only the family name has to mentioned.

-          The last sentence (rows 565-566) does not make sense in the content of the article ... probably it belongs to the Model paper.

Reviewer 3 Report

Comments and Suggestions for Authors

The content of the paper is relevant to the title. 

The abstract is short and concise. But I would definitely add some sentences describing the goal and importance of this paper. The abstract should be more compelling.

The literature review servers the purpose of this paper. However, I miss some clear goal statement in the introduction or abovementioned abstract. 

Methodology does not contain a clear research design. Why those selected methods were used?

The conclusion summarizes the results well, i appreciate the policy implication section. However, I would appreciate some discussion regarding works with similar focus -> how much your results differ from others scholars? Is there any space for further research? 

Can you more explain in the conclusion what it would practically mean that the threshold value would be crossed? Can it be explained with an example?

Regarding the implications. They are kind of general, so it would be good to show do what extent are these policies followed already or if there is an inclination that they are going to be followed?

Reviewer 4 Report

Comments and Suggestions for Authors

Dear authors, after reviewing this article, I present a set of suggestions with a view to improving the article.

Abstract – Simple, direct, well organized. Effectively summarizes the essence of the article, in accordance with the journal's guidelines

Introduction:

  – The first sentence of the introduction requires another framing (lines 23 and 24). Who says this? What is the context?

- The entire first paragraph contains a set of statements that appear as absolute truths, acquired data, without citation. Unless otherwise advised, this cannot happen in a scientific article, so authors must reformulate the entire initial part of the introduction.

- Line 36: “Agglomeration economy [1,2]” – I don’t understand, there must be something missing in this expression.

- Please highlight controversial and diverging hypotheses

- The authors should highlight the main conclusions.

Literature review:

The authors write that “The existing literature on green innovation efficiency, new energy industry and green finance is extensive and valuable.” However, the bibliography presented in this article is not abundant, on the contrary, the sources consulted are relatively scarce, and for the most part, a little old-fashioned. I recommend that authors make an effort to update their sources a little. The presentation of convergent and divergent theories would be useful for this purpose, would enrich the article and strengthen the relevance of the study.

In my opinion, it does not make much sense to present point 2 separately from point 3. These points should be united into one, with the title Literature Review and if it is later justified, subpoints can be opened.

Regarding the methodology and the empirical part, I believe this to be the strong point of this article. Congratulations on the quality of this section of the work.

Conclusions:

- “This section is not mandatory but can be added to the manuscript if the discussion is unusually long or complex.” – This expression should be removed.

- The authors must add the limitations of the present study and present clues for future investigations.

Round 2

Reviewer 1 Report

Comments and Suggestions for Authors

The article presents improvements compared to the previous version.

However, some bibliographic references are still not correct, (e.g.: page 114, 117, 119,…).

Reviewer 2 Report

Comments and Suggestions for Authors

In this new version of the manuscript, the authors carefully addressed our suggestions and observations made in the previous round or review, precisely explaining the changes made. Consequently, the work has significantly improved, being suitable for publication.

Reviewer 4 Report

Comments and Suggestions for Authors

Good job. The changes they made to the article greatly improved its overall quality.

Congratulations and good luck for future work
